# Wide range continuously tunable and fast thermal switching based on compressible graphene composite foams

Tingting Du[1,2,3], Zixin Xiong [1,3], Luis Delgado[1], Weizhi Liao[1], Joseph Peoples [1], Rajath Kantharaj[1], Prabudhya Roy Chowdhury [1], Amy Marconnet [1✉] & Xiulin Ruan [1✉]

Thermal switches have gained intense interest recently for enabling dynamic thermal management of electronic devices and batteries that need to function at dramatically varied ambient or operating conditions. However, current approaches have limitations such as the lack of continuous tunability, low switching ratio, low speed, and not being scalable. Here, a continuously tunable, wide-range, and fast thermal switching approach is proposed and demonstrated using compressible graphene composite foams. Large (~8x) continuous tuning of the thermal resistance is achieved from the uncompressed to the fully compressed state. Environmental chamber experiments show that our variable thermal resistor can precisely stabilize the operating temperature of a heat generating device while the ambient temperature varies continuously by ~10 °C or the heat generation rate varies by a factor of 2.7. This thermal device is promising for dynamic control of operating temperatures in battery thermal management, space conditioning, vehicle thermal comfort, and thermal energy storage.

[1] School of Mechanical Engineering and Birck Nanotechnology Center, Purdue University, West Lafayette, IN, USA. [2] School of Energy and Power Engineering, Shandong University, Jinan, Shandong, China. [3] These authors contributed equally: Tingting Du, Zixin Xiong. ✉email: marconnet@purdue.edu; ruan@purdue.edu

Thermal runaway and low-temperature performance degradation are major thermal issues for batteries in electronics, space applications, electric vehicles, and buildings and lead to serious safety risks and poor performance[1–4]. Depending on climate and operating conditions, batteries in these systems need effective heat dissipation pathways to dissipate the inherent heat generation or good thermal insulation (in low-temperature environment, for instance) to ensure reliable operation and optimal performance. However, thermal solutions beneficial across a range of operating conditions, such as at both high and low ambient temperatures, are still challenging[5]. Besides battery thermal management, other applications like space conditioning, vehicle thermal comfort, and thermal energy storage would benefit from controllable thermal conduction pathways. Switchable regulation of the heat dissipation pathway has the potential to achieve dynamic control of operating temperatures. A thermal switch, analogous to electrical switches that modulate current flow, tunes heat flux by changing thermal conductance of two-terminal components at "on" and "off" states. Beyond thermal switches with such sharp states transition, continuous and dynamic control of thermal conductance is desirable for thermal management applications and so-called "thermal regulators" need to be developed.

Past attempts to modulate thermal conductance focused on developing a variety of thermal switches and thermal regulators[6]. Gas-gap thermal switches physically make and break thermal contact between two components[7], which leads to a large ratio of the thermal conductance between the contacting (closed or "on" state) and non-contacting (open or "off" state) conditions[8, 9]. Liquid droplet-based thermal switches with droplets, actuated by an external field[10–13], jumping or moving across a micro-scale gap to contact the hot spot avoid mechanically moving parts. Thermal switches and thermal regulators based on phase change or phase transition leverage differences in thermal conductance between two states of a material[14–16], and a switching ratio of ~10x has been achieved[17]. Thermal switches utilizing electrochemical intercalation[18] or hydration[19] have also appeared recently. However, thermal switches do not have access to continuous tuning of the thermal conductance due to their binary "on" and "off" states. All solid-state thermal regulation has also attracted much attention recently. One type of solid-state thermal regulator exploits the effect of a thermal transport pathway at the boundary between metals and insulators to enable effective heat transfer only in one direction while the heat flow in the opposite direction is hindered, which makes these devices a type of thermal diode[6, 20–22]. All solid-state regulators are reliable and compact, but the switching ratio is low[18] and the devices can only operate in a narrow temperature window[23].

Here we propose and demonstrate a novel wide-range variable thermal resistor based on highly compressible, open-pore graphene foam composites, that can function as both thermal switches and thermal regulators as desired (schematically illustrated in Fig. 1). It is a thermal analogy to variable electrical resistor. When placed between a heat source and a heat sink, the foam modulates the allowed heat transport based on the degree of compression: when fully compressed, the foam conducts well ("on" state); while when fully uncompressed, it conducts heat poorly ("off" state). Unlike conventional thermal switches, with only "on" and "off" states, partially compressing the foam provides access to any intermediate thermal conductance. Further, the variable thermal resistor can be modulated with external active controls to constant uniform temperatures in varying environments and with varying heat loads. Here, we demonstrate a large tuning range of ~8x through measurements of the intrinsic thermal resistance of the material and the practical performance of temperature tuning and heat transfer regulation in an environmental chamber where the ambient temperature is varied from 0 °C to 30 °C. Cyclic tests further demonstrate the reliability of device.

## Results

### Our variable thermal resistor concept and the performance metrics

Thermal switches and thermal regulators are used to maintain devices at an optimal temperature during operation with varied heat load and ambient conditions. Thermal switches jump between the low and high thermal conductance states when activated, while thermal regulators have a non-linear relationship between heat flux and temperature which keeps a device at a constant temperature by modulating the heat flux (often passively). Schematically their operation is illustrated in Fig. 1a, b.

When considering such thermal devices, the ratio of heat flux to temperature difference across the device is defined as thermal conductance ($G$):

$$G = \frac{q''}{\Delta T} \tag{1}$$

For thermal switches, the thermal conductance is significantly different in the "on" and "off" states. As a figure of merit for switches, the switching ratio $r$ is defined as the ratio of these two thermal conductances:

$$r = \frac{G_{on}}{G_{off}} = \frac{q''_{on}/\Delta T_{on}}{q''_{off}/\Delta T_{off}}, \tag{2}$$

where $q''$ and $\Delta T$ are the heat flux and the temperature difference across the device and the subscripts indicate the "on" and "off" states. The higher the switching ratio, the more effective the system is as modulating the heat flux.

For thermal regulators, the relationship between heat flow and device temperature is non-linear, and the thermal conductance at a particular operating condition is defined as

$$G = \frac{\partial q''}{\partial \Delta T}. \tag{3}$$

Here the "on" and "off" states are passively determined by the current operating conditions and a switching ratio can be defined as a figure of merit as in Eq. (2).

In comparison to the traditional thermal switches and thermal regulators, our wide-range variable thermal resistor, shown in Fig. 1c, offers the benefits of large tuning range and continuous tuning simultaneously. It has the properties of a thermal switch with a minimum and maximum achievable thermal conductance in the "off" and "on" states, respectively, but can also achieve any intermediate state as highlighted in Fig. 1d. Thus, it can also function as a thermal regulator to maintain the device at a constant temperature in varying ambient conditions and with varying heat load. Therefore, in addition to the switching ratio, the available "temperature window" in which these devices can operate at a particular heat flux (horizontal, dashed purple line) and the "heat flux window" where the device temperature can remain constant (vertical dashed green line) are defined as performance metrics, as shown in Fig. 1d. Our approach can even realize any arbitrary $q''$-$T$ response curve as long as it is within the shaded region, opening possibilities for interesting applications.

### Thermal properties

We leverage composites consisting of commercially available graphene foams (Graphene Supermarket Products, Graphene/PDMS Flexible Foam) (Fig. 2) to achieve a variable thermal resistance. The graphene foam used in this work is grown by Chemical Vapor Deposition method with composition of 95% graphene and 5% PDMS. The foam thickness is 1.2 mm, and the density is 85 mg cm$^{-3}$. The graphene foam is reversibly compressible with macroscale open pores and interconnected networks of graphene as shown in Fig. 2. The pore size

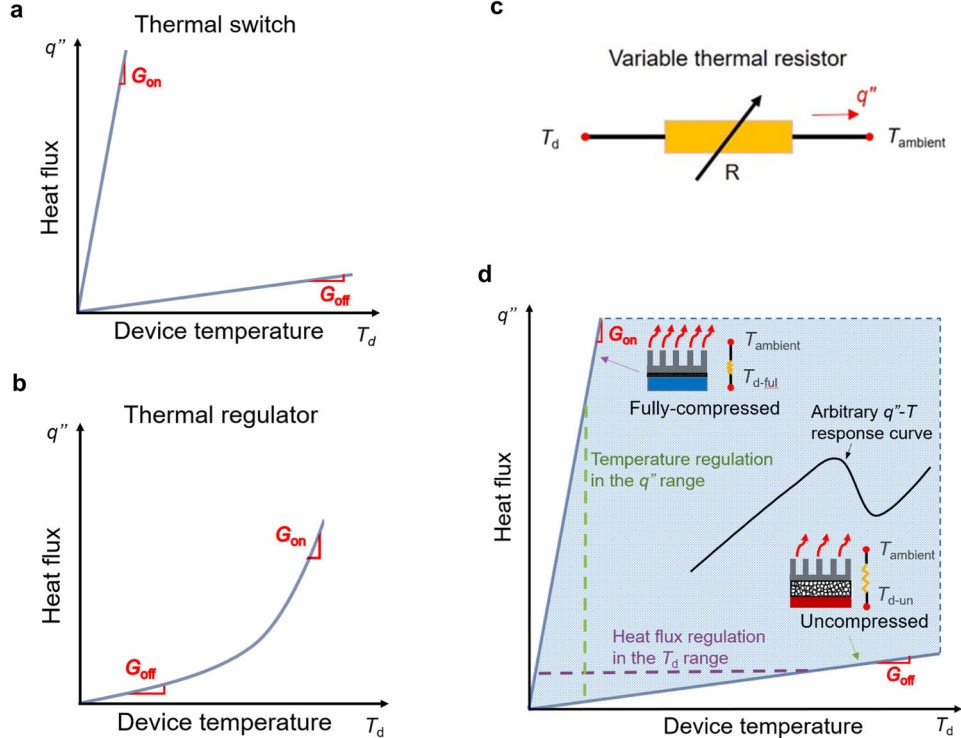

**Fig. 1 Working principle of thermal devices in tradition and in this work. a** A traditional thermal switch only has two possible thermal conductance states and **b** A traditional thermal regulator has a non-linear thermal performance curve and transitions from low to high thermal conductance passively based on device operating conditions. **c** Schematic of the variable thermal resistor concept. **d** Schematic showing the dual function of our device as both thermal switch and regulator as desired based on compressible graphene foam. The solid lines indicate the thermal performance in the "on" and "off" states (with the slopes being the thermal conductance of each state), and the shaded region indicates the region accessible with the continual tuning of the thermal conductance. As the power level or heat flux varies, the device temperature can be regulated through modulation of the foam thickness (along the dashed green, vertical line). Similarly, if a specific heat flux must be regulated, the thickness of the element can be controlled as the temperature varies (along the dashed purple, horizontal line).

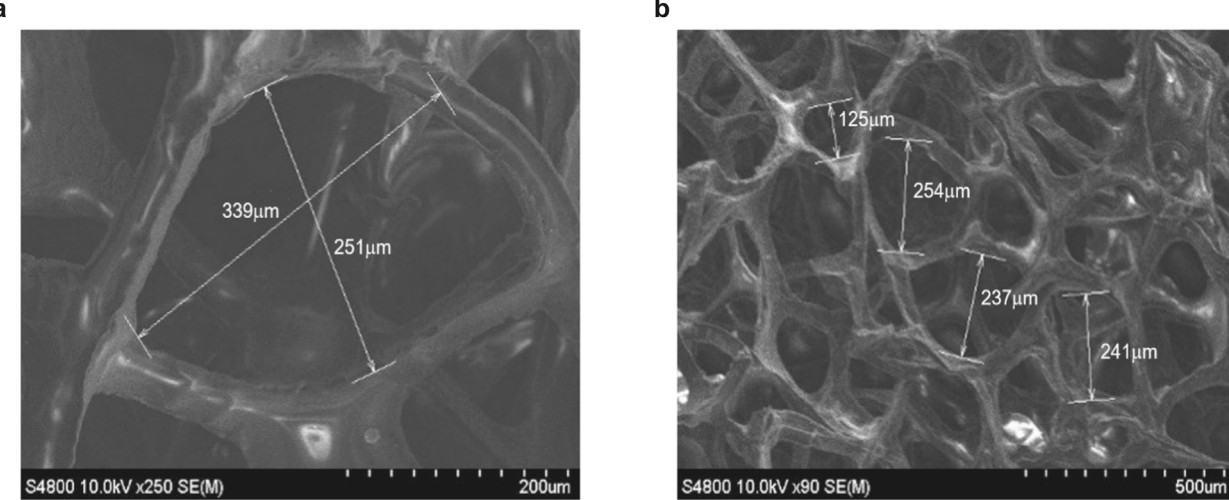

**Fig. 2 Images of the porous graphene foam by SEM. a** High magnification and **b** low magnification SEMs of the porous graphene foam illustrating the typical pore size of graphene foam and the interconnected network of graphene fabricated by chemical vapor deposition.

at the initial uncompressed state is measured between 150 and 350 µm by SEM images. The deformation of the pore structure during compression is shown in Supplementary Movie 1. The thermal conductance of graphene foam varies with the changing thickness. At the uncompressed state, the graphene foam provides good thermal insulation due to the low volume fraction of graphene and air within in pores. As the foam is compressed, the air

in pores is squeezed out of the compressed spaces and the pathways for heat transfer are more effective.

The thermal properties of the compressible graphene foam across a range of compression levels are first measured using a high-resolution infrared microscope (Quantum Focus Instruments (QFI) MWIR-1024 Infrascope) and a method based on the

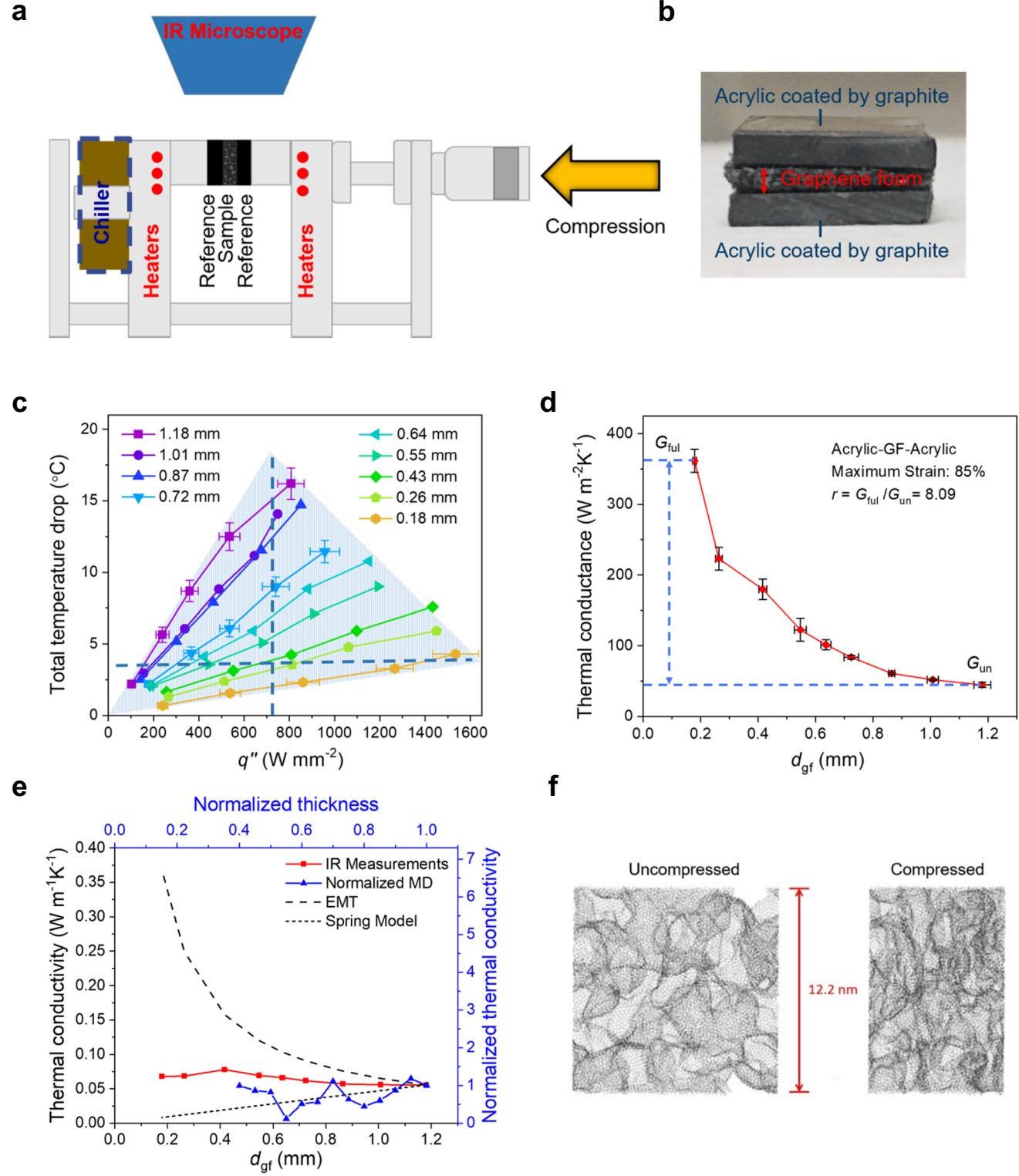

ASTM D5470 standard as illustrated in Fig. 3a, b and described in detail in the Methods section.

Briefly, at each compression level, a temperature gradient is established across the sample which is sandwiched between two reference regions of known thermal conductivity. The slope in the sample region and the total temperature drop at each thickness are extracted to understand the impact of contact thermal resistances. The heat flux through the sample is calculated based on the temperature gradient in the reference regions using Fourier's Law and neglecting convection losses (see Methods section for details). In this experiment, for each compression level, temperature maps are recorded at six heat fluxes (~100–1600 kW m$^{-2}$). The average thickness of compressed graphene foam during each test is measured from the thermal images.

Figure 3c shows that the total temperature drop at each thickness increases approximately linearly with increasing heat flux, and that

**Fig. 3 Thermal property characterization setup, results, and MD simulation domain. a** Schematic of IR Reference Bar Setup. The graphene foam is sandwiched between two layers of material with known thermal conductivity (i.e., the reference layers) and the three-layer structure is placed between two adapter bars that span from a heat source to the heat sink during property testing. Both sides have embedded cartridge heaters to achieve the uniform temperature required for calibration of emissivity. During test the heaters on the hot side and the chiller on the cold side are used to establish a temperature gradient across the reference-sample-reference stack. The IR microscope measures the 2D temperature map of the top surface of the sample stack. The graphene foam is compressed by moving the caliper. The emissivity is calibrated, and a new temperature map is captured by IR microscope after the compression to measure properties as function of compression. **b** Image of the three-layer sandwich structure with the graphene foam in the middle and the acrylic of known thermal conductivity on two sides. The commercially available acrylic is coated by graphite to enhance the emissivity during IR measurements. **c** The total temperature drops from the hot to cold side of the foam at each thickness as a function of heat flux. Representative error bars are shown for the minimum, the maximum, and an intermediate thickness. The shaded region illustrates the tunable range of temperature and heat flux via continuous compression. The dashed lines indicate the ability to stabilize the heat flux and the temperature in an allowed temperature range and an allowed heat flux range, respectively. **d** The thermal conductance as a function of compression. Here, $d_{gf}$ is the thickness of graphene foam. The switching ratio between the fully compressed and uncompressed states is ~8.09 at the maximum strain of ~85%. **e** Measured thermal conductivity and scaled MD simulation results compared with prediction from the EMT and our proposed spring model. The IR microscope measurement is bounded by the EMT and the proposed spring model. **f** The side view of simulation domain in MD simulations of black carbon atoms on white background at uncompressed (left) and compressed (right) states.

the temperature rise increases with increasing thickness. The shaded region illustrates the tunable temperature range available during the compression. The blue dashed lines illustrate that the temperature and the heat flux can be fixed at a desired value while the heat flux and temperature, respectively, are allowed to vary if the foam thickness is adjusted appropriately. At a given thickness, the slope of the total temperature drop across the sample with heat flux ($\Delta T\, q''^{-1}$) is the total thermal resistance across the sample, which is inversely proportional to the thermal conductance (shown in Fig. 3d). Compared to the traditional thermal switches which only have a high and low value, this graphene foam can achieve continuous tuning of thermal conductance smoothly between the fully compressed and uncompressed states. The minimum thickness of the sample at fully compressed state may vary across compression systems. The apparatus shown in Fig. 3a is not designed to apply large forces, and "fully compressed" here refers to the smallest thickness that can be achieved on this setup. At a strain of ~85%, the switching ratio based on the ratio of the thermal conductance at the fully compressed and uncompressed state is ~8.09.

The measured thermal conductivity of the foam is illustrated in Fig. 3e and shows intriguing behavior: it increases with increasing mass density (decreasing thickness), but much more slowly as compared to the conventional effective medium theory (EMT) for porous media[24], where the thermal conductivity increases almost linearly with mass density (i.e., thermal conductivity is inversely proportional to thickness) as more thermal pathways are available within a unit volume after compression. To better elucidate the microscopic mechanism behind the dynamic thermal transport, we predict the thermal conduction in compressed foams using molecular dynamics (MD) simulations. Fig. 3f shows the side view of the cubic simulation domain before and after compression, visualized using VMD[25]. The foam in MD is much scaled down from the actual foam used in the experiment but provides useful and relevant insights. The thermal conductivity at each compression level is calculated using the Green–Kubo method[26, 27], the details of which are discussed in the Methods section. The results, normalized with respect to the uncompressed state, are plotted in Fig. 3e on the secondary axis (the corresponding thermal conductivities from experiment are plotted on the primary axis). It should be noted that the predicted thermal conductivity has considerable level of oscillation due to the flexible nature of the foam. Interestingly, the MD calculated thermal conductivity initially decreases and then increases with increasing mass density (decreasing thickness). To explain this unexpected initial decreasing trend, we propose a 1D spring model with variable thickness x between the hot and cold reservoirs. Details of the spring model are discussed in the Supplementary Note 1 and Supplementary Fig. 1. Briefly, the effective thermal conductance ($k_{eff}\, x^{-1}$) across

the spring remains constant with varying x, since heat travels the same distance through the coils of the spring wire (assuming adjacent spring coils do not come in contact). Therefore, the effective thermal conductivity ($k_{eff}$) decreases with decreasing thickness ($x$), which explains the initial compression trend in MD. Once the foam is compressed more and the ligaments start to make contacts, the thermal conductivity starts to increase, as revealed by the MD simulation. The experimental data are now bounded between the EMT and the spring model, indicating that both mechanisms influence the thermal behavior of the sample under compressive loading. The MD results initially follow the spring model as compression occurs and then show features of the EMT. Overall, this comparison reveals that the dependence of thermal conductivity on mass density due to compression is distinctive from that due to initial growth density for porous media.

**Environmental testing**. To evaluate the performance of our variable thermal resistor, a proof-of-concept experiment is conducted in an environmental chamber with varying ambient temperature. The experimental setup is illustrated in Fig. 4 and described in more detail in the Methods section.

To test the temperature regulation capacity of the graphene foam, the input heat flux is fixed at 3023 W m$^{-2}$ and the ambient temperature is adjusted to 0 °C, 10 °C, 20 °C, and 30 °C. The graphene foam is first set at the uncompressed ("off") state, and then compressed by 10% of the original thickness each time until the maximum compression is achieved (the "on" state). At each thickness of the graphene foam, the device system is allowed to achieve steady state.

For a constant heat input, Fig. 5a shows the device temperature as a function of ambient temperature for different thicknesses of the foam. The approximately linear relationship between the ambient temperature and the device temperature at the same thickness of graphene foam indicates no major non-linearity in properties or performance with temperature. The average temperature difference between the device and ambient temperatures are 17.4 °C and 7.0 °C for the uncompressed and fully compressed states, respectively, yielding a switching ratio of ~2.5. The switching ratio is likely lower in application than in the thermal property testing because of the other parallel heat loss pathways in this system, but it can be optimized.

The same data is illustrated in a contour plot in Fig. 5b showing that the device temperature can be continuously tuned across a temperature window of ~10 °C (illustrated by the blue dashed line in Fig. 5a) at this input heat flux for each ambient temperature condition. To maintain the device at a constant temperature with varying ambient conditions, the graphene foam thickness $d_{gf}$ should

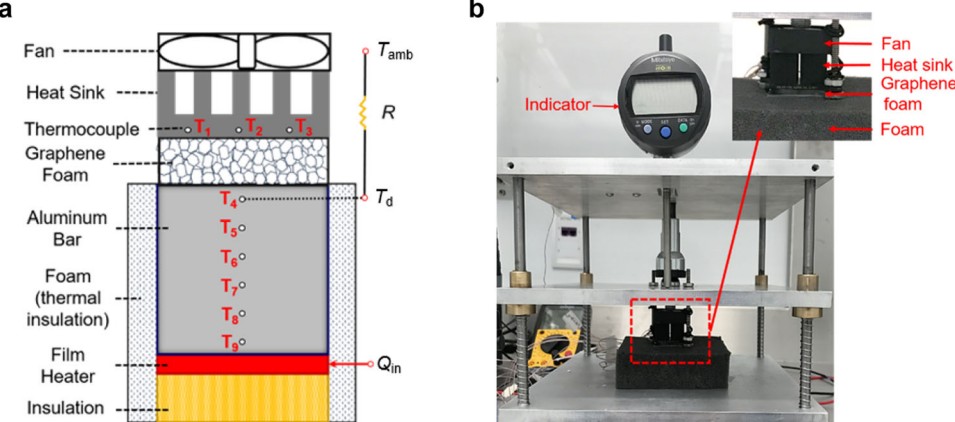

**Fig. 4 Setup of the proof-of-concept experiment in an environmental chamber. a** Schematic and **b** Photo of the performance test set-up. A film heater is used to mimic a device and placed on a block of hard polymer foam insulation to minimize heat dissipation through the baseplate. An aluminum bar (1″ × 1″ in cross-section and 0.8″ tall) is placed above the heater with 6 thermocouples to quantify the heat flow. The thickness of the graphene foam, inserted between the aluminum bar and the heat sink, can be controlled by moving the middle plate shown in panel **b**. An indicator measures the movement of the middle plate and is related to the thickness of graphene foam. The temperature of the top thermocouple ($T_4$), which is 0.1″ from the surface, is taken as the device temperature for the calculation of thermal regulation to remove the effect of the added aluminum bar. Three thermocouples additional are distributed within the base of heat sink. Another thermocouple, floating in the air, measures the ambient temperature. The whole system is contained within an environmental chamber where the ambient temperature can be adjusted from 0° to 30 °C.

be tuned by following the desired isotherm. If the desired temperature is not a fixed point, but a range, such as from 20 °C to 30 °C, the temperature window over which the device can operate expands to 20.4 °C (displayed by the blue shade and the range between the brown and blue dashed lines in Fig. 5a).

To further explore the operational performance of these continuously tunable thermal regulators, we next explore the impact of varying input power to our "device" corresponding to heat fluxes from ~1.57 kW m$^{-2}$ to ~6.05 kW m$^{-2}$ when the ambient temperature is held at 20 °C (see Fig. 5c, d). The device temperature increases as the heat flux increases and temperature difference between the uncompressed and fully compressed states is larger for the higher heat flux than that in lower heat flux cases as the temperature difference from the device to ambient ($T_d$ − $T_{amb}$) scales approximately linearly with heat flux. The contour plot illustrates that device temperature can also be maintained at a desired value by adjusting the thickness of graphene foam as the heat flow varies. When the device temperature is kept at 30 °C, the heat flux window can vary by ~3 kW m$^{-2}$, or a factor of ~2.7 (displayed by the brown dashed line in Fig. 5c).

**Reliability, cycling, and time constant**. To investigate the reliability of graphene foam-based variable thermal resistor in temperature regulation, the system is cycled between the fully compressed and uncompressed states for 10 cycles (see Fig. 6a). The temperature difference between the system and the environment demonstrates an average system switch ratio of 2.9 with a standard error of 2%. We can note that the switching of the thermal resistance of our approach is nearly instant, but it does take time for the temperature distribution to reach a new steady state. Fig. 6b shows the evolution of the device temperature during cycling. It takes 22.1 min to raise the device temperature by ~11.2 °C when transitioning from the fully compressed ("on") state to the uncompressed ("off") state. In contrast, the time spent in cooling the device after compressing again is only 10.7 min. This contrast in the off-on and on-off transition is logical as the system time constant should scale with the thermal resistance of the graphene foam. For the uncompressed state, the foam has higher thermal resistance and a longer system time constant. For the compressed state, the resistance is lower and the time constant reduced. Further, the ability to quickly cool when

compressing might aid in preventing thermal runaway or system damage. The time constant could be significantly shortened if the aluminum bar is reduced or removed. It is needed in our prototype measurements using thermocouples but can be removed in actual applications. The full transient temperature response of 10 continuous cycles is shown in Supplementary Note 2 and Supplementary Fig. 2, where the time constants can be seen to be stable. Further, we have measured the stress-strain relation in compress-release cycle of a representative composite foam. The results are discussed in Supplementary Note 3 and Supplementary Fig. 3, which again confirm the robustness of the foam and small hysteresis of the mechanical properties.

To summarize, we demonstrate all solid-state, wide-range continuously tunable, and fast thermal switching based on compressible graphene composite foams, which are more versatile than conventional thermal switches that only possess two discrete "on" and "off" states. Characterization of the thermal properties of graphene foam demonstrate that the temperature and heat flux can be both tuned with dual functionality of thermal switching and thermal regulation. The switching ratio at the maximum compression of 85% is ~8. For a heat flux of ~3 kW m$^{-2}$, the adjustable temperature window is ~10 °C between the fully compressed and uncompressed state, enabling stabilization of a device temperature during operation in varying ambient conditions. Meanwhile, the reliability and reasonable response time permit the compressible graphene foam to be an optimal all solid-state candidate for dynamic thermal management. Based on the uniformity in thermal properties and thermal behavior across a number of samples used in the tests, this thermal switching system is expected to show similar performance on a larger scale.

## Methods
**Thermal property characterization**. A high-resolution IR microscopy (Quantum Focus Instruments (QFI) MWIR-1024 Infrascope) is used to measure the thermal conductance of graphene foam at different thicknesses. Fig. 3 illustrates the system, and Fig. 7a shows a schematic of the test section with the direction of heat flow and compression. In this setup during thermal characterization, the hot side bar is heated by cartridge heaters and the cold side bar remains at a relatively steady temperature by liquid-cooled heat sink. The graphene foam is placed between two acrylic layers of known thermal conductivity with the same cross-section area. The surface of acrylic is spray-coated with a thin layer of graphite (CRC Products, Dry Graphite Lube) to achieve uniform emissivity and to decrease the uncertainty in the temperature

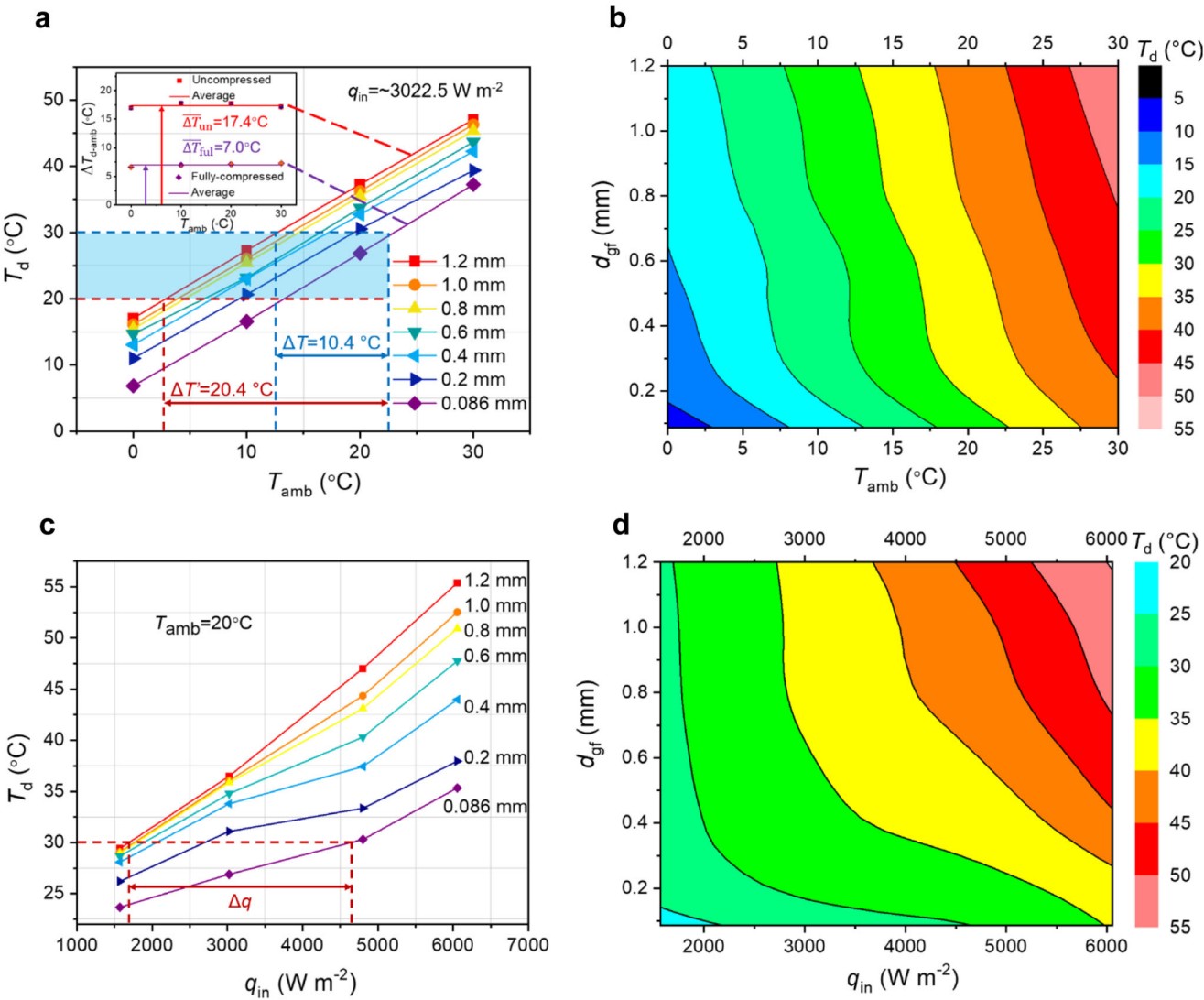

**Fig. 5 Performance as both a thermal regulator and a thermal switch. a** Device temperature as a function of ambient temperature at each thickness of graphene foam, for a constant heat flux. The temperature difference between device and environment at the uncompressed and fully compressed states are 17.4 °C and 7.0 °C, respectively, resulting in a switching ratio of ~2.5. The temperature window over which the device temperature can be controlled is ~10 °C from the minimum to maximum $T_d$ at each ambient temperature level at this heat flux (the range illustrated by the blue dashed line) and can be extended to 20.4 °C (the range between the brown and the blue dashed lines) when the operating temperature is a range from 20 °C to 30 °C (displayed by the blue shade). **b** Contour plot showing the capacity of graphene foam for continuous temperature regulation with varying ambient temperatures. To achieve a constant device temperature, the thickness of the graphene foam, $d_{gf}$, should be adjusted following one of the isotherms as the ambient temperature is varied. **c** Device temperature as a function of the heat flux at different thicknesses of the graphene foam. **d** Contour plot showing the capacity of graphene foam for continuous regulation based on heat load. To maintain a particular device temperature above the ambient temperature, the thickness of the graphene foam can be controlled following the isotherms as the input heat flux is varied.

measurements. To calibrate the spatially varying emissivity, a radiance image is taken after the sample is heated to a known and uniform temperature. Then the emissivities of graphene foam and the reference layers are calculated at each pixel by comparing their radiance to that of a blackbody at the reference temperature. After calibration, a temperature gradient is applied across the sample stack and the calculated emissivity map is used to calculate the temperature distribution. The 2-D temperature map of surface is captured with a spatial resolution of 5 μm/pixel and a temperature resolution of ~0.1 K.

The 1-D temperature profile is calculated by averaging the 2-D temperature map in the direction perpendicular to the heat flow at the steady state (see Fig. 7b). Based on Fourier's Law, the heat flux is calculated from the temperature gradient in the reference and the thermal conductivity of the acrylic (which is previously verified against gum rubber, a Standard Reference Materials certified by the National Institute of Standards and Technology (NIST)). The total temperature drop across the sample can be identified from the temperature profile including the temperature jump at the left and right interface if significant. The thermal conductance of graphene foam, $G$, can be

expressed as follows:

$$G = \frac{1}{R''} = \frac{k_{\text{ref}} \frac{dT}{dx}\big|_{\text{ref}}}{\Delta T}. \tag{4}$$

For each thickness of graphene foam, the temperature maps are captured at five heater power levels. The thickness of graphene foam is measured as the distance between the two interfaces identified in the temperature profile. At the uncompressed state, a small pressure is required to hold the sample in place. Thus, the initial thickness is slightly smaller than the sample outside of the measurement system. The minimum compressible thickness varies depending on the maximum load on the entire stack. Fig. 7b shows the temperature profile at the uncompressed and fully compressed state at the same input power. After compression, the temperature difference between two interfaces decreases, which means the thermal conductance of graphene foam has increased. The ratio of thermal conductance at the fully compressed and uncompressed state is taken as the figure of merit

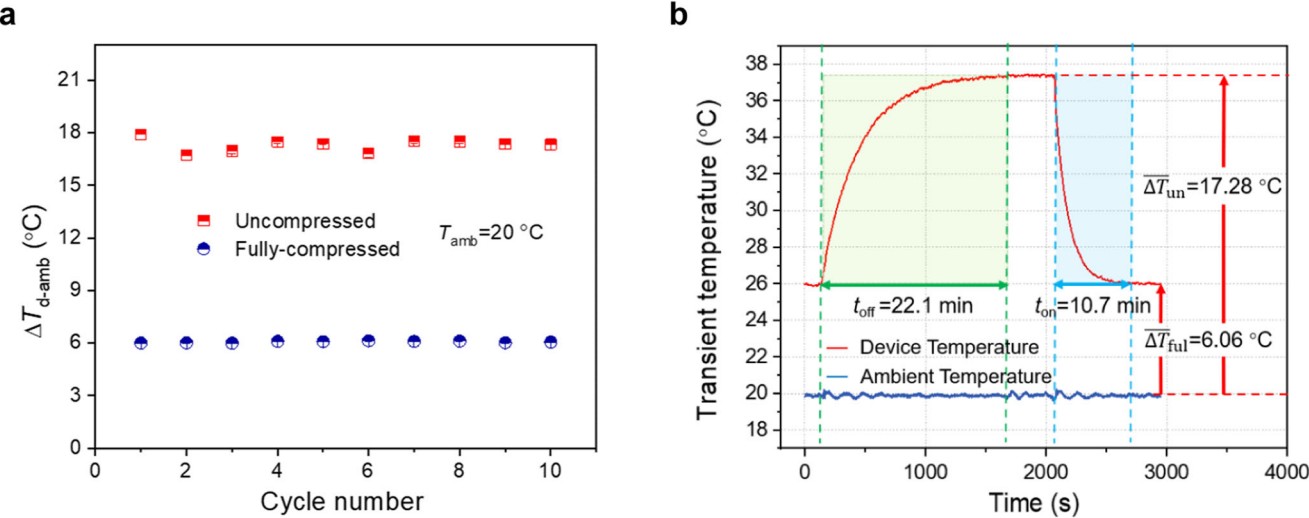

**Fig. 6 Performance on reliability and response time. a** Device-ambient temperature difference ($\Delta T_{\text{d-amb}}$) at the uncompressed and fully compressed states during cycling. The graphene foam is stable with very small temperature deviation of each test point. **b** Example transient temperature response for the system during 1 cycle of compression. The time to steady state when transitioning from the "on" (compressed) to "off" (uncompressed) state is ~22 min, while transitioning from "off" to "on" is ~11 min.

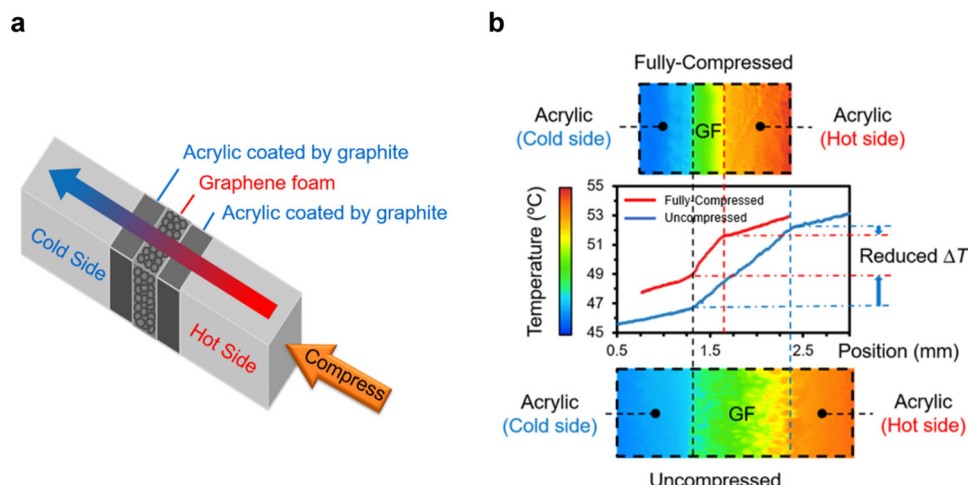

**Fig. 7 Thermal property characterization and uncertainty quantification. a** Schematic of sample stack illustrating the direction of heat flow and compression. **b** 2-D temperature maps for the fully uncompressed (bottom) and fully compressed (top) states and the corresponding 1-D temperature profiles. The reduced temperature difference across the sample indicates an increase of thermal conductance after compression. As illustrated in the 1-D profiles (i.e., the lack of a temperature jump at the interfaces), the interfacial resistance between the sample and the reference layers is negligible compared to the thermal resistance of the sample itself.

(Eq. (5)):

$$r = \frac{G_{\text{fully-compressed}}}{G_{\text{uncompressed}}}. \tag{5}$$

**Uncertainty quantification**. The uncertainty in $G$ stems from the uncertainty in the measured heat flux and the uncertainty in the measured total temperature drop between two interfaces of graphene foam. According to the Fourier's law, the uncertainty in the measured heat flux is due to the uncertainty in the thermal conductivity of reference (which is 10% of reported value) and in the temperature gradient in the reference regions. In the IR temperature measurements, the accuracy of temperature gradient is influenced by the accuracy of T-type thermocouple which is used to measure the reference temperature $T_{\text{ref}}$ for the emissivity calibration. Another factor impacting the uncertainty in the reference temperature gradient is the manual selection of the reference regions. The uncertainty of total temperature drop across the sample is also impacted by the manual selection of the region with the foam. Specifically, it is quantified by shifting the manually selected reference-sample interface by several pixels. Ultimately, the combined uncertainty is evaluated by the uncertainty propagation approach. The combined uncertainty in the thermal conductance of graphene foam with three compression levels is represented by the error bars shown in Fig. 3d in

Results section and is ~9.50% for the 1.18 mm thickness, ~9.32% for the 0.72 mm thickness, and ~8.93% for the 0.18 mm thickness.

**Molecular dynamics simulation**. The graphene foam model (shown in Fig. 3f) is the same as that used in the work of Pedrielli et al.[28] and was provided to us by the authors. We use the Tersoff potential[29] to describe the interatomic forces and a timestep of 0.5 fs. Periodic boundary conditions are applied in all directions. By remapping atom positions, the system is compressed to desired levels (the compression process is shown in the Supplementary Movie 2), followed by an equilibration period in a canonical (NVT) ensemble at 300 K to eliminate instability introduced when applying compressive strain. The thermal conductivity at each compression level is calculated using the Green–Kubo method[26, 27]:

$$k = \frac{V}{3 k_{\text{B}} T^2} \int_0^\infty \langle \vec{\mathbf{J}}(0) \cdot \vec{\mathbf{J}}(t) \rangle \, dt. \tag{6}$$

where the heat current $\vec{\mathbf{J}}$ is:

$$\vec{\mathbf{J}} = \frac{1}{V} \left( \sum_i E_i \vec{\mathbf{v}}_i + \frac{1}{2} \sum_{i<j} (\vec{\mathbf{F}}_{ij} \cdot (\vec{\mathbf{v}}_i + \vec{\mathbf{v}}_j) \vec{\mathbf{r}}_{ij}) \right). \tag{7}$$

Here, $V$ is the volume of the system, $E$ is the total energy per atom, $\mathbf{v}$ is the velocity

of each atom, **F** is the pair-wise force between atoms $i$ and $j$ obtained using the interatomic potential, and **r** is the distance between two atoms. The MD simulations are performed using LAMMPS[30].

**Environmental testing**. The system thermal regulation performance of graphene foam is tested in an environmental chamber with a controlled ambient temperature. The entire experimental setup is shown in Fig. 4. A film electrical heater, clamped between a thermal insulator and an aluminum bar, supplies the desired amount of heat. The input power is tuned by controlling the applied power and calibrated by a multimeter. The graphene foam is placed between the aluminum bar and the heat sink. It is compressed by the heat sink which is connected to the movable middle plate. The distance of movement (related to the compressed thickness of graphene foam) is measured by an indicator with accuracy of ±0.0001 in. T-type thermocouples, coated with heat-conducting epoxy, are inserted into the holes of the aluminum bar and the heat sink. Another thermocouple floats in the air to ensure the temperature adjustment of the chamber is steady. The temperature variation with the thickness and heat flux is recorded by a data logger with an accuracy of ±1.7 °C. The temperature of the upper thermocouple in the bar is taken as the device temperature. The graphene foam and the heat sink with a fan contribute to the total resistance in the system. The average thermal resistance of the heat sink with fan is separately measured at 1.64 K W$^{-1}$, which is 4.8% and 39.6% of the graphene foam resistance at the uncompressed and fully compressed states, respectively.

In the performance experiment with the variation of heat flux, the ambient temperature is fixed at 20 °C, and the heat flux based on the electrical power is approximately 6.05 kW m$^{-2}$. After the system approaches steady state, the graphene foam is compressed by 10% of its thickness and the system is allowed to reach a new steady state. This process is repeated until the foam is compressed to its reachable minimum thickness. Then, the sample is expanded to its original thickness and the power is reduced and the process is repeated at electrical power-based heat fluxes of ~4.80 kW m$^{-2}$, ~3.02 kW m$^{-2}$, and ~1.57 kW m$^{-2}$.

In the performance experiment with varying ambient temperature, the input heat flux is fixed at ~3.02 kW m$^{-2}$, and the ambient temperature for the test is stabilized at 0 °C, 10 °C, 20 °C, and 30 °C. In order to reduce the measurement time and uncertainty, the ambient temperature cycles among the four set points at each thickness of graphene foam.

As the compression is manually adjusted, it is difficult to achieve exactly the same thickness repeatedly. Small elastic deformation of each component in the setup also induces some uncertainty in the sample thickness. Further, in this experiment, due to the heat loss by convection, radiation, and conduction through the insulation to the surroundings, the true amount of heat flowing through the graphene foam is smaller than the input heat flux. Thus, the switching ratio achieved in the system performance tests is smaller than in the material property testing where the exact heat flux through the sample is measured and these losses are negligible.

## Data availability
The source data of Figs. 3, 5, 6, and Supplementary Figs. 2 and 3 are provided in the Supplementary Data, and further request should be directed to the corresponding authors. Source data are provided with this paper.

## Code availability
LAMMPS molecular dynamics package is available at https://lammps.sandia.gov/. The custom codes used in this work are available from the corresponding authors upon reasonable request.

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

## Acknowledgements
T.D. acknowledges the support from the China Scholarship Council (CSC). L.D. acknowledges the Carver Fellowship from Purdue University.

## Author contributions
X.R. and A.M. conceived the experiments and supervised the work. T.D., L.D., and Z.X. conducted the thermal property characterization experiment by IR reference bar method and analyzed all temperature maps. T.D. and R.K. calculated the uncertainty in thermal conductance in the thermal property characterization experiment. Z.X. performed the modeling studies. T.D., W.L, and J.P. designed the proof-of-concept setup and conducted the experiment in the environmental chamber. R.K. measured the stress-strain relation. T.D., Z.X., A.M. and X.R. co-wrote the paper. All authors discussed the results and commented on the manuscript.

## Competing interests
X.R., A.M., T.D., and L.D. are the inventors of a provisional patent application with US Serial No. 63/083,303 filed on September 25, 2020.
