## [Peer Review File · Nature Communications]

REVIEWER COMMENTS

Reviewer #1 (Remarks to the Author):

- 1) The authors designed a new thermal switch/regulator using the graphene foam, and characterize the thermal switch properties, reliability, cycling and time constant. In addition, the several uncertainties were also analyzed.
- 2) The setup type of the high resolution infrared microscope is suggested to be presented.
- 3) The details of the emissivity calibration should be given.
- 4) There are a couple of typos should be corrected, e.g., Line 12 & 13, two approaches; Line 126, the unit of density is wrong.

Reviewer #2 (Remarks to the Author):

The manuscript from Du et al. entitled "Wide-Range Continuously Tunable, Fast, and Scalable Thermal Switching Based on Compressible Graphene Composite Foams" proposed the thermal device for dynamic control as a thermal regulator. The authors demonstrated the performance of the thermal regulator device, which was made by the composite material between graphene and PDMS. According to the results and discussion, I have some comments that can be addressed in a revised version of the manuscript below.

- It will be meaningful if the authors explain how to prepare and characterize the graphene composite foam in this manuscript. The authors should also compare the optimized method with others reported in terms of the percentages of graphene in PDMS elastomer.
- The authors should provide the mechanical properties of the graphene composite foam, such as stress, strain, or Young's modulus at the critical heat flux during fully compressed and uncompressed.
- In Fig. 3c, the authors should describe the overlapped data between 1.008 mm and 0.865 mm thick of the graphene composite foam.
- It would be a benefit of this manuscript if the authors can provide the hysteresis data of heat flux and temperature difference.
- The authors need to describe the thermal regulator behaviors of the device, such as the distribution of the graphene in PDMS or the interconnection network of graphene inside the composite material. It would be better if the authors can provide or propose the thermal conduction model of the graphene composite foams in micro or nanoscale.
- The authors should avoid using a.k.a. in scientific writing.

Overall, I found this manuscript to be reasonably well written, but I have a substantial concern about its suitability for the journal and its lack of criticism of the thermal regulator mechanism in micro or nanoscale. As a result, I cannot support publication in the present form.

Reviewer #3 (Remarks to the Author):

The authors report the used of graphene composite form to achieve a tunable, fast and scalable thermal switch. Within the manuscript, the authors demonstrated that by adjusting the thickness of the graphene form, heat flux (across the heat source and heat sink) can be varied accordingly. Additionally, the authors also demonstrated the reliability of the concept via performing the cyclical "on" and "off" state of the thermal switch by compressing and uncompressing the graphene form. This study demonstrates an interesting application of the graphene form. My comments as follow:

1. As stated in the title (and abstract), the proposed thermal switch is scalable. However, within the

manuscript, there was no demonstration on this. Would the proposed thermal switch show similar performance on a smaller scale system?

2. How is the "fully compressed state" is defined? In Fig. 3(c), the under the fully compressed state, the thickness of the form was 0.179 mm, whereas in Fig. 5, the thickness was 0.086 mm. Thus, the "fully compressed state" is depended on the system to squeeze the form?

3. Although in Fig. 6(a) the data showed the temperature difference was relatively constant under 10 times of "on" and "off" state, what about the porous structure of the form? Would there any damage to the porous graphene form structure under extended compression?

4. Under compression, any variation in thickness of the graphene form across the entire sample area?

5. Since Fig. 6(b) showing the transient temperature response for the system for one-cycle under "on" and "off" state, would this transient response remained the same in the 1st and that on the 10th cycle?

REVIEWER COMMENTS

Reviewer #1 (Remarks to the Author):

1) The authors designed a new thermal switch/regulator using the graphene foam, and characterize the thermal switch properties, reliability, cycling and time constant. In addition, the several uncertainties were also analyzed.

2) The setup type of the high resolution infrared microscope is suggested to be presented.

Response: We used a Quantum Focus Instruments (QFI) MWIR-1024 Infrascop (infrared microscope) to characterize thermal properties of the foam sample. This information has been added to the first sentence of the second paragraph in Thermal Properties section in Results:

“...measured using a high-resolution infrared microscope (Quantum Focus Instruments (QFI) MWIR-1024 Infrascop) and a method...”

3) The details of the emissivity calibration should be given.

Response: We thank the reviewer for the helpful comment. The following statement has been added to Thermal Property Characterization in Methods section:

“To calibrate the spatially varying emissivity, a radiance image is taken after the sample is heated to a known and uniform temperature. Then the emissivities of graphene foam and the reference layers are calculated at each pixel by comparing their radiance to that of a blackbody at the reference temperature. After calibration, a temperature gradient is applied across the sample stack and the calculated emissivity map is used to calculate the temperature distribution.”

4) There are a couple of typos should be corrected, e.g., Line 12 & 13, two approaches; Line 126, the unit of density is wrong.

Response: Thank you. The typos have been corrected.

Reviewer #2 (Remarks to the Author):

The manuscript from Du et al. entitled “Wide-Range Continuously Tunable, Fast, and Scalable Thermal Switching Based on Compressible Graphene Composite Foams” proposed the thermal device for dynamic control as a thermal regulator. The authors demonstrated the performance of the thermal regulator device, which was made by the composite material between graphene and PDMS. According to the results and discussion, I have some comments that can be addressed in a revised version of the manuscript below.

- It will be meaningful if the authors explain how to prepare and characterize the graphene composite

foam in this manuscript. The authors should also compare the optimized method with others reported in terms of the percentages of graphene in PDMS elastomer.

Response: We thank the reviewer for the helpful advice. In this research, the graphene foam we used was commercially purchased from the Graphene Supermarket Products. The method of preparing PDMS is relatively mature ([Z Chen et al., *Nat. Mater.* **10**, 424-428 (2011)], [Z Chen et al., *Adv. Mater.* **25**, 1296-1300 (2013)], and [B. H. Min et al., *Carbon* **80**, 446-452 (2014)]) and the properties of the material are supplied by the Graphene Supermarket Products (<https://graphene-supermarket.com/Graphene-PDMS-Foam.html>). Thus, in our original manuscript, we cited the manufacturer and gave all information on preparation, composition, and characteristics of the foam we have from the manufacturer in these sentences:

“Here we leverage composites consisting of commercially-available graphene foams (Graphene Supermarket Products, Graphene/PDMS Flexible Foam) to achieve variable thermal resistance. The graphene foam is grown by Chemical Vapor Deposition method with composition of 95% graphene and 5% PDMS. The foam thickness is 1.2 mm, and the density is 85 mg cm^{-3} .”

- The authors should provide the mechanical properties of the graphene composite foam, such as stress, strain, or Young’s modulus at the critical heat flux during fully compressed and uncompressed.

Response: We have measured the stress-strain relation of the sample and added the following in the main text:

“Further, we have measured the stress-strain relation in compress-release cycle of a representative composite foam. The results are discussed in Supplementary Note 3 and Supplementary Fig. 3, which again confirm the robustness of the foam and small hysteresis of the mechanical properties.”

We also added the following as the Supplementary Note 3:

“The stress-strain relation of the foam sample has been measured using Instron mechanical tester (E-1000 series). We recently purchased a new batch of samples from the same vendor and used it for stress-strain testing. Some characteristics of the new sample differ slightly from that of the old one in terms of thickness and maximum compressive strain. Potentially, the fabrication process from the vendor has certain variation. However, as the composition of the foam remains the same, we believe the mechanical test results of the new foam can represent the typical mechanical behaviors of the previously used one. For mechanical testing, the sample is compressed uniaxially from 1.467 mm to 0.734 mm (~50% compressive strain). Load is applied and released at constant velocity of 0.01 mm s^{-1} and some hysteresis observed on compression versus release. In Supplementary Fig. 3, the stress in the release path is greater than 0 when reaching 0 strain, meaning the sample fully recovers to its original thickness from 50% compressive strain. The two paths differ slightly with reduced stress at the same strain level in release path. Similar mechanical behavior was reported in literature including for graphene grown with a CVD method on nickel foam then coated with PDMS [Z Chen et al., *Adv. Mater.* **25**, 1296-1300 (2013)], a 3D graphene network grown on porous ceramic SiO_2 substrate [H Huang et al., *J. Mater. Chem. A* **2**, 18215-18218 (2014)], and a 3D graphene formed in pyrrole-containing graphene oxide suspension [Y Zhao et al., *Adv. Mater.* **25**, 591-595 (2013)].”

- In Fig. 3c, the authors should describe the overlapped data between 1.008 mm and 0.865 mm thick of the graphene composite foam.

Response: We attribute the cause of the overlapped data in Fig. 3c to uncertainties in both measured heat flux and measured total temperature drop. The heat flux is obtained based on thermal conductivity of

reference material and temperature gradient in reference regions. The reported thermal conductivity of reference has uncertainty of 10%. The uncertainty in temperature gradient is caused and influenced by the accuracy of T-type thermocouple used in reference temperature measurement for emissivity calibration and the manual selection of reference region in data analysis. Details of uncertainty quantification is covered in Method section, and similar uncertainty (~10%) is found in cases with different thicknesses. Representative error bars for three data sets were plotted in Fig. 7c in our original manuscript and have now been moved into Fig. 3c. We have removed Fig. 7c and added the following sentence in description of Fig. 3:

“Representative error bars are shown for the minimum, the maximum, and an intermediate thickness.”

- It would be a benefit of this manuscript if the authors can provide the hysteresis data of heat flux and temperature difference.

Response: We have measured the stress-strain relation in compress-release cycle of the recently purchased new foam sample, and the result has been plotted in Supplementary Fig. 3. As discussed in our previous response, the measured stress in release path is slightly less than that in compress path, which has been reported in literature on graphene foams.

The transient temperature response during one compress-release cycle is illustrated in Fig. 6b. The resistance of the sample changes nearly instantaneously, but the temperature distribution takes time to reach new steady state. The time required for off-on transition (i.e., compress path) is 10.7 minutes, and 22.1 minutes for on-off transition (i.e., release path). The time constant scales with thermal resistance of the foam: at on state, resistance is low and temperature response is fast; while at off state, high resistance leads to slower temperature response. We have added the following in the main text:

“The full temperature response of 10 continuous cycles is shown in Supplementary Note 2 and Supplementary Fig. 2, where the time constants can be seen to be stable.”

We have also added the transient response in 10 continuous cycles as the Supplementary Note 2. The time constant in both path across 10 cycles is stable.

- The authors need to describe the thermal regulator behaviors of the device, such as the distribution of the graphene in PDMS or the interconnection network of graphene inside the composite material. It would be better if the authors can provide or propose the thermal conduction model of the graphene composite foams in micro or nanoscale.

Response: We thank the reviewer for the comment. It is a 3D interconnected graphene foam, and a small amount (5%) of PDMS is coated on the graphene ligaments. The structure is shown in Fig. 2. We have now performed thermal conductivity modeling, and added the following in the results section:

“The measured thermal conductivity of the foam is illustrated in Fig. 3e and shows intriguing behavior: it increases with increasing mass density (decreasing thickness), but much more slowly as compared to the conventional effective medium theory (EMT) for porous media²⁴, where the thermal conductivity increases almost linearly with mass density (i.e., thermal conductivity is inversely proportional to thickness) as more thermal pathways are available within a unit volume after compression. To better elucidate the microscopic mechanism behind the dynamic thermal transport, we predict the thermal conduction in compressed foams using molecular dynamics (MD) simulations. Figure 3f shows the side view of the cubic simulation domain before and after compression, visualized using VMD²⁵. The foam in MD is much scaled down from the actual foam used in the experiment but provides useful and relevant insights. The thermal conductivity at each compression level is calculated using the Green-Kubo^{26,27}, the details of which are discussed in the

Methods section. The results, normalized with respect to the uncompressed state, are plotted in Fig. 3e on the secondary axis (the corresponding thermal conductivities from experiment are plotted on the primary axis). It should be noted that the predicted thermal conductivity has considerable level of oscillation due to the flexible nature of the foam. Interestingly, the MD calculated thermal conductivity initially decreases and then increases with increasing mass density (decreasing thickness). To explain this unexpected initial decreasing trend, we propose a 1D spring model with variable thickness x between the hot and cold reservoirs. Details of the spring model are discussed in the Supplementary Note 1. Briefly, the effective thermal conductance ($k_{\text{eff}}x^{-1}$) across the spring remains constant with varying x , since heat travels the same distance through the coils of the spring wire (assuming adjacent spring coils do not come in contact). Therefore, the effective thermal conductivity (k_{eff}) decreases with decreasing thickness (x), which explains the initial compression trend in MD. Once the foam is compressed more and the ligaments start to make contacts, the thermal conductivity starts to increase, as revealed by the MD simulation. The experimental data are now bounded between the EMT and the spring model, indicating that both mechanisms influence the thermal behavior of the sample under compressive loading. The MD results initially follow the spring model as compression occurs and then show features of EMT. Overall, this comparison reveals that the dependence of thermal conductivity on mass density due to compression is distinctive from that due to initial growth density for porous media.”

- The authors should avoid using a.k.a. in scientific writing.

Response: The acronym has been removed from the last sentence in Fig.7 description.

Overall, I found this manuscript to be reasonably well written, but I have a substantial concern about its suitability for the journal and its lack of criticism of the thermal regulator mechanism in micro or nanoscale. As a result, I cannot support publication in the present form.

Response: We appreciate the reviewer’s time and effort at reviewing our work. The constructive comments aided to improving our manuscript as documented in this rebuttal document. We hope the revised version has adequately addressed the concern.

Reviewer #3 (Remarks to the Author):

The authors report the used of graphene composite form to achieve a tunable, fast and scalable thermal switch. Within the manuscript, the authors demonstrated that by adjusting the thickness of the graphene form, heat flux (across the heat source and heat sink) can be varied accordingly. Additionally, the authors also demonstrated the reliability of the concept via performing the cyclical “on” and “off” state of the thermal switch by compressing and uncompressing the graphene form. This is study demonstrates an interesting application of the graphene form. My comments as follow:

1. As stated in the title (and abstract), the proposed thermal switch is scalable. However, within the manuscript, there was no demonstration on this. Would the proposed thermal switch show similar performance on a smaller scale system?

Response: We thank the reviewer for the helpful comments. The graphene/PDMS foam samples used in thermal properties and environmental testing in this work have dimensions of 1cm × 1cm and 1” × 1”, respectively. Similar thermal properties and behavior are expected to be observed in samples with larger

dimensions. Hence, the proposed thermal switch would show similar performance on a larger scale system. To avoid confusion, “scalable” has been removed from the title, and the following sentences have been added to the abstract and conclusion sections:

“Based on the uniformity in thermal properties and thermal behavior across a number of samples used in the tests, this thermal switching system is expected to show similar performance on a larger scale.”

2. How is the “fully compressed state” is defined? In Fig. 3(c), the under the fully compressed state, the thickness of the form was 0.179 mm, whereas in Fig. 5, the thickness was 0.086 mm. Thus, the “fully compressed state” is depended on the system to squeeze the form?

Response: The minimum thickness the foam can achieve may vary across different compressing systems due to different designs and amount of pressure that can be applied by each system. The apparatus under IR microscope used in this work is able to provide axial loading to the sample. The maximum available force, however, is limited, compared to what our environmental chamber apparatus can offer. Hence, the reported thickness at “fully compressed state” is different in Fig.3c and Fig.5. We have revised our definition of “fully compressed state” in Fig.3c to be the “fully compressed state achieved in the IR apparatus”.

We have added the following sentences in Results section:

“The minimum thickness of the sample at fully compressed state may vary across compression systems. The apparatus shown in Fig. 3a is not designed to apply large force, and “fully compressed” here refers to the smallest thickness that can be achieved on this setup.”

3. Although in Fig. 6(a) the data showed the temperature difference was relatively constant under 10 times of “on” and “off” state, what about the porous structure of the form? Would there any damage to the porous graphene form structure under extended compression?

Response: We have measured the stress-strain relation of recently purchased foam across 3 cycles (Supplementary Fig. 3) as described in a previous response. In the release path, the measured stress remains above 0 before returning to 0 strain position, which means the foam recovers to its original thickness completely. We believe it indicates that there is little damage and deformation in the foam structure after cycles. Besides, the relatively constant temperature difference among 10 cycles in Fig. 6a also indirectly shows small changes in structure, since thermal properties highly depend on the foam structure.

4. Under compression, any variation in thickness of the graphene form across the entire sample area?

Response: In experiments on both the IR measurement apparatus and the environmental chamber setup, we observed very little change in thickness across the entire sample area. The foam did not extend beyond the edges of reference layers and aluminum bar, as shown in Fig. 3a and Fig. 4a respectively.

5. Since Fig. 6(b) showing the transient temperature response for the system for one-cycle under “on” and “off” state, would this transient response remained the same in the 1st and that on the 10th cycle?

Response: We have plotted the transient temperature variation for 10 cycles in the supplementary information of the revised manuscript. As seen in the Supplementary Fig. 2, the temperature at each uncompressed state and compressed state at nearly 0.086 mm thickness are similar over 10 cycles. The time required for temperature to rise/drop by 10 °C are 3.87 ± 0.63 minutes and 1.21 ± 0.25 minutes,

respectively. Although experimental error cannot be avoided in the compression process, it can be demonstrated that the cycling performance of graphene foam is stable as evident in the figure and by the 2% standard error bar of switch ratio which is discussed in the Methods section.

We have added the following paragraph as the Supplementary Note 2:

“We measured the evolution of temperature difference across the sample during 10 cycle on environmental chamber setup between uncompressed state and fully compressed state at 0.086 mm. The time required for the temperature to rise/drop by 10 °C are 3.87 ± 0.63 minutes and 1.21 ± 0.25 minutes, respectively. Although experimental error cannot be avoided in the compression process, it can be demonstrated that the cycling performance of graphene foam is stable as evident in Supplementary Fig. 2.”

REVIEWERS' COMMENTS

Reviewer #1 (Remarks to the Author):

The authors have answered qll my questions, the manuscript can be accepted for publication in the revised form.

Reviewer #2 (Remarks to the Author):

The manuscript from Du et al. entitled "Wide-Range Continuously Tunable, Fast, and Scalable Thermal Switching Based on Compressible Graphene Composite Foams" proposed the thermal device for dynamic control as a thermal regulator. The author has improved and added information according to the suggestions given as well. Therefore, I would like to support this document to be published.

Reviewer #3 (Remarks to the Author):

The responses from the authors on my comments are satisfactory.

REVIEWER COMMENTS

Reviewer #1 (Remarks to the Author):

The authors have answered all my questions, the manuscript can be accepted for publication in the revised form.

Reviewer #2 (Remarks to the Author):

The manuscript from Du et al. entitled “Wide-Range Continuously Tunable, Fast, and Scalable Thermal Switching Based on Compressible Graphene Composite Foams” proposed the thermal device for dynamic control as a thermal regulator. The author has improved and added information according to the suggestions given as well. Therefore, I would like to support this document to be published.

Reviewer #3 (Remarks to the Author):

The responses from the authors on my comments are satisfactory.

Response to all reviewers: Thank you very much for the positive recommendation!